# Timing of Entry into Paid Employment, Adverse Physical Work Exposures and Health: The Young Helsinki Health Study

**DOI:** 10.3390/ijerph17217854

**Published:** 2020-10-27

**Authors:** Tea Lallukka, Rahman Shiri, Olli Pietiläinen, Johanna Kausto, Hilla Sumanen, Jaana I. Halonen, Eero Lahelma, Ossi Rahkonen, Minna Mänty, Anne Kouvonen

**Affiliations:** 1Department of Public Health, University of Helsinki, P.O. Box 20, 00014 Helsinki, Finland; olli.k.pietilainen@helsinki.fi (O.P.); hilla.sumanen@helsinki.fi (H.S.); eero.lahelma@helsinki.fi (E.L.); ossi.rahkonen@helsinki.fi (O.R.); minna.manty@helsinki.fi (M.M.); 2Finnish Institute of Occupational Health, P.O. Box 18, 00032 Helsinki, Finland; rahman.shiri@ttl.fi (R.S.); johanna.kausto@ttl.fi (J.K.); 3Department of Health Care and Emergency Care, South Eastern Finland University of Applied Sciences, 48220 Kotka, Finland; 4Department of Health Security, Finnish Institute for Health and Welfare, P.O. Box 30, 00271 Helsinki, Finland; jaana.halonen@thl.fi; 5Department of Strategy and Research, City of Vantaa, 01030 Vantaa, Finland; 6Faculty of Social Sciences, University of Helsinki, 00014 Helsinki, Finland; anne.kouvonen@helsinki.fi; 7Research Institute of Psychology, SWPS University of Social Sciences and Humanities, 53-238 Wroclaw, Poland

**Keywords:** occupational cohort, social determinants, working conditions, health behaviors, obesity, self-rated health, young employees

## Abstract

It is not well known how the timing of entry into paid employment and physical work exposures contribute to different health outcomes in young employees. Thus, we determined the associations of age at entry into paid employment and physical work exposures with general and mental health in young employees and determined whether associations differ by behavior-related risk factors. Data were collected via online and mailed surveys in autumn 2017 from employees of the City of Helsinki aged 18–39 years (n = 5897; 4630 women and 1267 men, response rate 51.5%). Surveys comprised measures of age at entry into paid employment, seven working conditions, behavior-related risk factors and health outcomes (self-rated health [SRH] and common mental disorders [CMD] as generic indicators of physical and mental health). Logistic regression analysis was used. After full adjustment, age at entry was not associated with the health outcomes; however, in additional analyses, younger age at first employment was associated with smoking and obesity (OR 3.00, 95% CI 2.34–3.85 and 1.67, 95% CI 1.32–2.11 for those started working at age of ≤18 years, respectively). Of the working conditions, sitting and standing were positively associated with poor SRH and CMD and uncomfortable working postures with CMD. Working conditions were broadly similarly associated with health outcomes among those with and without behavior-related risk factors. Although we found little support for modification by behavior-related risk factors, overweight, obesity and smoking were associated with poor SRH and binge drinking and smoking with CMD. Additionally, moderate and high levels of leisure-time physical activity were inversely associated with poor SRH. In conclusion, early entry into paid employment appears not to associate to immediate poorer health in young employees, although it was associated with smoking and obesity even after full adjustment. Exposure to physically heavy work and uncomfortable working postures may increase the risk of adverse health outcomes.

## 1. Introduction

Entering paid employment is an important step in young people’s lives that gives them not only financial security, but also other types of stability on their path to adulthood, whereas forms of non-employment such as unemployment are linked to different adverse mental and physical health problems [1,2,3,4] and even to an increased risk of work disability due to common mental disorders [5]. However, it is unclear whether age at entry into paid employment makes a difference and whether there are sensitive periods which could be associated with more health problems among young employees. Furthermore, early entry into paid employment is likely to be linked to physically heavier work through shorter education.

Earlier studies have shown that physically demanding work—typically present in lower occupational class jobs—is associated with poorer health [6,7]. Adverse work exposures are also associated with behavior-related risk factors [8,9,10], which can further independently contribute to subsequent health problems. However, many of these previous studies have considered midlife and aging employees, or they have not distinguished between younger and older employees. Thus, it is not clear if the associations exist already during early working life span. If so, the detection and prevention of early risk factors are needed for extending work careers.

It is likely that socioeconomic background, e.g., low parental socioeconomic position and low education, play a role in both age at entry and the type of work in terms of its physical exposures. Accordingly, a previous study using birth cohort data from Northern Finland showed that low childhood socioeconomic position and unhealthy behaviors were associated with young age at entry into first paid employment, particularly among those who had obtained high education themselves [11]. Similar factors also predicted having a low occupational class or no registered occupation in the first job. Late entry into paid employment, in turn, could be assumed to reflect higher education, although heterogeneous patterns with different effects such as having children should not be ruled out. It is also possible that those who enter early but later obtain higher education can often be from socioeconomically more disadvantaged backgrounds, and they need to start to work earlier to finance their education as compared to those from more affluent backgrounds. These assumptions were supported by our earlier study, where participant’s educational level shaped the associations between social and health related predictors and entry into paid employment [11]. In all, life course social determinants and behavior-related risk factors should be considered when examining the associations between timing of entry, physical work exposures and health outcomes during early work careers.

If early entry into paid employment is linked to more physical work, it could be assumed to manifest as health problems later on, but also later entry could be an indicator of health issues. Indeed, timing of entry into paid employment is potentially linked to a wide range of health-related outcomes including work (non-)participation, as a proxy indicator of health. A birth cohort study showed that those who had their first period of paid employment at an older age had higher odds for long-term unemployment over the follow-up [12]. Regarding behavior-related risk factors, smoking is more common among those who begin their work careers early, as compared to those who have acquired higher education and have therefore delayed entry into paid employment or starting a family [13].

Our assumption was that those young adults who entered paid employment earlier, as compared to their counterparts who, e.g., began their working life after a longer education, could be more exposed to adverse working conditions that are assumed to affect their health. This could subsequently affect their work participation in the long run. It is further hypothesized that there could be sensitive periods in early adulthood when the effects of adverse working conditions are the most harmful. This alongside physical work exposures could also be reflected as behavior-related risk factors shaping physical and mental health of young employees.

Therefore, to confirm the significance of both timing of entry and physical work exposures to established indicators of general health, this study had two aims: (1) to examine whether timing of entry into paid employment and physical work exposures are associated with self-rated health and common mental disorders; and (2) to determine whether behavior-related risk factors modify the associations, i.e., are the examined associations different or similar between individuals with or without behavior-related risk factors. Additional analyses further showed the results stratified by gender and considered behavior-related risk factors as separate outcomes.

## 2. Materials and Methods

### 2.1. Study Design and Participants

Data for this study were collected in autumn 2017 among all employees of the City of Helsinki who were born in 1978 or later and had been employed for at least 4 months (typical probation period) with a contract of 50% or more [14]. Survey data were collected via secure online server, and the questionnaires were mailed to those without email addresses or who did not respond via the online link. Additional data were collected during telephone interviews among non-respondents to online and mailed surveys. The final response rate was 51.5%, and the non-response analysis shows that, despite the relatively low response rate, the data are broadly representative of the target population with respect to key variables of interest, i.e., sociodemographic and work-related factors and health [14]. For the purposes of this study, we included survey data from online and mailed survey questionnaires only (88% of the final respondents), since the telephone interview did not comprise all of the variables needed for this study. Both online and mailed questionnaires were similar and comprised measures on entry into paid employment, sociodemographic and migrant background, working conditions, behavior-related risk factors and health. The final number of participants in the online and mailed surveys to be included in this study was 5897 (4630 women and 1267 men). The gender distribution matches that of municipal employees in the target population [14] and in municipal employees in general. Thus, in the target population, the proportion of women was 77%, and that of men 23%, while the corresponding figures among respondents were 79% and 21%, respectively. In turn, of all Finnish municipal employees, 80% were women in 2018 (https://www.kt.fi/en/municipal-sector-and-personnel).

The ethical committee of the Faculty of Medicine, University of Helsinki gave their approval for the study plan. Additionally, the City of Helsinki provided permission to conduct this study. All participants were given detailed information about the study, and then they could choose if they wanted to participate and return the survey.

### 2.2. Measures

#### 2.2.1. Predictors: Timing of Entry into Paid Employment and Physical Work Exposures

Entry into paid employment was based on age when the participants reported they had their first actual job, excluding summer jobs during education and other shorter training periods. Broadly following previous procedures [11,12], and reflecting a typical age to finish higher education (by 23 years) in Finland (for more details of the Finnish educational system, see e.g., https://minedu.fi/en/education-system), age at entry was classified into four groups distinguishing between early, average and late entry into first paid employment: ≤18 years, 19–21 years, 22–24 years (reference) and ≥25 years, respectively.

Physical work exposures were assessed using an 18-item inventory on physical working conditions and sedentary work [15]. We focused on seven different physical work exposures: (1) heavy physical exertion or lifting and carrying heavy loads; (2) uncomfortable working postures; (3) trunk rotation; (4) repetitive movements; (5) sitting; (6) standing; and (7) vibration. Each item had four response alternatives: (1) No, does not occur; (2) Yes, they occur, but they pose no problems at all; (3) Yes, they occur, and they do pose problems to a certain extent; and (4) Yes, they occur, and they pose a major problem. These variables were dichotomized as “No” (1 and 2) and “Yes” (3 and 4).

#### 2.2.2. Health Outcomes

Health outcomes included self-rated health (SRH) and common mental disorders (CMD) (measured by general health questionnaire GHQ-12) as key indicators of general and mental health. SRH was measured by a single item from the short-form 36 (SF-36) questionnaire [16,17]. The item asked the participants to report if they perceived their current health in general as poor, moderate, good, very good or excellent. Those reporting their health as moderate or poor scored 1, while others served as a reference. The measure has been widely used in epidemiological studies, and it has been shown to be a strong predictor of other, more objective adverse outcomes such as morbidity, disability pension and mortality [18,19,20].

GHQ-12 is a validated and reliable inventory which captures symptoms of usually minor mental health problems. However, GHQ-12 is also predictive of more severe mental disorders [21,22,23,24]. In general, CMD indicate context-free affective, non-psychotic mental health problems, typically depressive and anxiety symptoms, or self-esteem problems. There are 12 items, each scoring 1 point (range 0–12), and the time frame to assess symptoms in the previous few weeks. We used a recommended cut off point of 3 or more symptoms to indicate common mental disorders.

Additional analysis considered a more specific health indicator, low back pain (LBP), which is prevalent already in young adults and in many different populations [25,26,27]. It was assessed by three questions: (1) Are you suffering from any pains or aches right now (no, yes)? (2) When did the pain begin (up to 3 months ago, more than 3 months ago)? (3) Where do you feel the pain (neck or shoulders, lower back, one or both upper extremities, one or both lower extremities, face or elsewhere in head, elsewhere)? LBP was defined as pain in the lower back for less than or more than three months. Chronic LBP was defined as pain lasting longer than three months.

#### 2.2.3. Potential Modifiers of the Associations: Health Behaviors and Obesity

We included health behaviors and overweight/obesity both as potential modifiers to the examined associations between work exposures and health, as covariates (see the next section), and as separate outcomes in supplementary analyses. In tables and results, they are jointly referred to as behavior-related risk factors. Smoking was classified as never smoker, past smoker, occasional smoker or current smoker, when it was used as a predictor. It was dichotomized when it was used as an outcome, comparing current smokers with other participants. Due to small number of smokers (Table 1), smoking could not be used as a potential modifier. The numbers for binge drinking also were not sufficient to do the subgroups analyses, thus it was used as covariate. The duration and intensity of both leisure-time and commuting related activities were assessed and a metabolic equivalent (MET) value was calculated for each participant [28]. Leisure-time physical activity was first dichotomized into inactive and active, using the median value of MET as cut off point. The dichotomized variable for leisure time physical activity was used as on outcome and for stratification. When physical activity was used as a predictor, tertiles of MET were used.

Body mass index (BMI, kg/m^2^) was based on self-reported weight and height, and it was classified as underweight (BMI under 18.5), healthy weight (BMI 18.5–24.9), overweight (BMI 25.0–29.9) and obesity (BMI 30 or more). As an outcome, it was dichotomized to distinguish participants with obesity from others. In the stratified analyses, it was dichotomized to distinguish participants with overweight or obesity from others because small numbers of people with obesity did not allow for stratified analyses.

#### 2.2.4. Covariates

As covariates, we included gender, age, migrant background (Finland vs. other), participant’s education, household income, father’s education, mother’s education, BMI, smoking, physical activity and binge drinking. As indicators of socioeconomic position, participant’s own education (elementary, vocational, upper secondary, bachelor’s degree, master’s degree or doctoral degree), his/her parental education (elementary, vocational, upper secondary or higher education) and household income (10 categories) were included. Age was used as a continuous variable in the analyses. Alcohol consumption was based on frequency of binge drinking (drinking six units or more on a single occasion). The item comprised four response alternatives: never, less frequently than once a month, once a month or once a week or more frequently.

Additionally, psychosocial work exposures (job demands and job control) were considered as covariates. Job demands and job control were measured with Karasek’s job content questionnaire, using a shorter version as in previous studies [29,30]. Both scores were dichotomized to indicate low and high job demands and low and high job control, to create a job strain variable (high demands coupled with low job control, compared to other combinations).

### 2.3. Statistical Analysis

Logistic regression models were fitted to examine the associations of timing of entry into paid employment and physical work exposures with health outcomes. First, we ran models adjusting for age and gender or adjusting for age in gender-specific analysis. Second, we ran full multivariable models adjusting for all covariates. We also performed stratified analyses to determine whether leisure-time physical activity and overweight or obesity modify the associations between physical work exposures and health-related outcomes. Gender-specific results are shown in the Appendix A. The study design and the assumed relationships between the variables are illustrated in a simplified schematic Figure 1. We used STATA, version 15 (StataCorp LLC, College Station, TX, USA) for the analyses.

## 3. Results

### 3.1. Characteristics of the Study Population

Mean age of the participants was 32 years (SD 4.6 years). About a fourth of the participants had their first entry into paid employment when they were 22–24 years old (Table 1). It was less common to begin working at the age of 18 or younger, but still 17% had entered paid employment before they turned 19. In contrast, nearly a third of participants had their first job when they were 25 years or older. Reporting different types of physical work exposures was common among participants (Table 1). There were differences in the nature and prevalence of the physical work exposures, with exposures such as awkward or uncomfortable working postures being particularly prevalent (56% reported such exposures), whereas vibration was the least prevalent exposure, reported by only 5% of the participants.

Mean BMI was 25.2 kg/m^2^ (SD 5.0), and 14% of the participants were obese (Table 1). Of the participants, 6% drank six units or more on a single occasion at least once a week, 11% were current smokers, 12% were occasional smokers and 20% had stopped smoking. The prevalence of poor self-rated health (SRH) was 11% and that of CMD 35%.

### 3.2. Health-Related Outcomes

Table 2 shows age- and gender-adjusted odds ratios for the associations of age at entry into paid employment, physical work exposures and behavior-related risk factors with health outcomes. Compared to those who had entered paid employment between ages 22 and 24, those who had entered between ages 19 and 21 were somewhat more likely to report poor SRH (OR 1.26, 95% 1.01–1.57) and CMD (OR 1.23, 95% CI 1.05–1.44). Heavy physical exertion or lifting and carrying heavy loads, uncomfortable working postures, trunk rotation, repetitive movements, sitting and standing were all associated with poor SRH and CMD.

Table 2 also shows the results of multivariable models for the associations of gender, migrant background, behavior-related risk factors and physical work exposures with health outcomes., Sitting (OR 1.67, 95% CI 1.33–2.09) and standing (OR 1.69, 95% CI 1.31–2.18) were associated with poor SRH. Uncomfortable working postures (OR 1.22 95% CI 1.02–1.45), sitting (OR 1.25 95% CI 1.08–1.44) and standing (OR 1.69 95% CI 1.41–2.03) were associated with reporting CMD. Current smoking, overweight and obesity were positively associated with poor SRH, while moderate and high levels of leisure-time physical activity were inversely associated with poor SRH in multivariable models. Smoking (past and current) and binge drinking once a month or more often were positively associated with CMD.

### 3.3. Behavior-Related Risk Factors as Potential Modifiers of the Associations

Next, we determined whether leisure-time physical activity and overweight or obesity could modify the associations between physical work exposures and health-related outcomes. In the presence of behavior-related risk factors in the models, the goodness-of-fit of the model was poor. We therefore did not adjust for behavior-related risk factors. Most of the associations between physical work exposures and poor SRH and CMD did not differ between participants with healthy weight and overweight or obesity (Table 3), or between participants with low or high level of leisure-time physical activity (Table 4). An association between uncomfortable working postures and poor SRH was suggested among employees with overweight/obesity (OR = 1.41, 95% CI 0.99–1.99, P = 0.051) but not among employees with healthy weight. Standing was associated with poor SRH in employees with low level of leisure-time physical activity (OR = 1.90, 95% CI 1.45–2.49) but not in those with high level of leisure-time physical activity. Furthermore, uncomfortable working postures were associated with CMD only in employees with low level of leisure-time physical activity (OR = 1.23, CI 1.01–1.51), and repetitive movements were associated with CMD only in employees with high level of leisure-time physical activity (OR = 1.36, CI 1.01–1.84).

### 3.4. Additional Analyses

The results were relatively similar among women and men (Appendix A). Binge drinking once a week or more often was positively associated with poor SRH among women. Among men, such association could not be confirmed (Appendix A).

Early entry into paid employment and physical exertion or lifting and carrying heavy loads were not associated with LBP (Appendix A). Uncomfortable working postures (OR = 1.21, 95% CI 1.04–1.41), current smoking (OR 1.30, 95% CI 1.05–1.61) and being overweight (OR 1.18, 95% CI 1.01–1.39) or obese (OR 1.24, 95% CI 1.01–1.52) were associated with LBP after adjustment for age and gender. However, after further adjustment for other confounding factors, only uncomfortable working postures (OR = 1.28, 95% CI 1.03–1.58) were associated with LBP.

The associations between physical work exposures and LBP did not significantly differ between participants with healthy weight, overweight or obesity or between participants with low or high level of leisure-time physical activity (Appendix A). Uncomfortable working postures and repetitive movements were associated with LBP participants only among participants with healthy weight.

Compared with participants who had entered paid employment between ages 22 and 24, smoking was 3.00 (95% CI 2.34–3.85) times and obesity 1.67 (95% CI 1.32–2.11) times more likely among participants who entered paid employment at age 18 years or younger (Appendix A). Among those who entered paid employment between ages 19 and 21, smoking was 2.04 (95% CI 1.62–2.58) times and obesity 1.84 (95% CI 1.50–2.26) times more likely. Obesity and smoking were less likely in participants who had entered paid employment at age 25 or older.

Participants who were exposed to physical exertion or lifting and carrying heavy loads or spent more hours on physically demanding work per day were more likely to smoke or be with obesity and were more likely to exercise during leisure-time than those who were not exposed. Uncomfortable working postures, trunk rotation, repetitive movements and standing were positively associated with poor SRH and CMD, while sitting was inversely associated with poor SRH and leisure-time physical activity.

The multivariable analysis showed that participants who had entered their first paid employment at age 18 or younger (OR 1.96, 95% CI 1.43–2.70) or at age 25 or older (OR 1.48, 95% CI 1.02–2.13) had higher odds of regular smoking, as compared to those who had entered paid employment between ages 22 and 24 (Appendix A). Additionally, those who had entered their first paid employment between ages 19 and 21 were more likely to be with obesity (OR 1.49, 95% CI 1.17–1.89). Age at first employment was unassociated with physical activity.

The number of hours of physically demanding work per day was positively associated with daily smoking, whereas exposure to vibration was inversely associated with daily smoking. Uncomfortable working postures and standing at work were positively associated with obesity. While the exposure to standing at work was associated with lower leisure-time physical activity, the number of hours of physically demanding work per day was positively associated with leisure-time physical activity.

Regarding the associations among behavior-related risk factors, the frequency of binge drinking was particularly strongly associated with smoking (OR 12.86, 95% CI 8.36–19.79 for binge drinking at least once a week). Moderate and high levels of leisure time physical activity were inversely associated with daily smoking. Past smoking was positively associated with obesity, while moderate and high levels of leisure-time physical activity were inversely associated with obesity. Current smoking, overweight or obesity and binge drinking for at least once a week were associated with lower leisure-time physical activity.

## 4. Discussion

### 4.1. Main Findings

This study sought to determine the associations of timing of entry into paid employment and physical work exposures with health outcomes during early work careers. The main results show that early entry into paid employment is not associated with poorer health in young employees. The additional analyses, however, highlighted that smoking and obesity were more likely among employees who started their work careers early. In turn, physical work exposures were associated with self-rated and mental health particularly before full adjustments and sitting and standing after full adjustments. Furthermore, most of the associations between physical work exposures and poor SRH and CMD were mostly similar among employees with and without behavior-related risk factors, as well as among women and men.

### 4.2. Interpretation

Our results are broadly in line with previous studies focusing on entry into paid employment, working conditions and health-related factors [11,13,31,32,33]. However, early entry into paid employment has rarely been used as a predictor of health or health-related outcomes. Among these young employees, we did not observe differences in associations with SRH or CMD. A study showed an association between smoking and early entry [11], but early entry has seldom been used as a predictor of health. As our study is cross-sectional, our findings could be assumed to reflect similar patterns, as smoking likely has been initiated already before working life. Nonetheless, we cannot determine the causal order between the predictor (early entry into paid employment) and the outcome (e.g., smoking). Physical working conditions have been associated with SRH and CMD [34,35,36] in older employees, but few studies have before confirmed their significance to these health outcomes already during earlier and mid careers. As physical work is still prevalent, these results highlight the importance of early detection of risk groups to promote health and prevent the health problems from becoming chronic. Moreover, since SRH and CMD are both also predictors of subsequent work disability [18,37,38,39], this emphasizes the need to focus on early prevention and potential modification of their risk factors such as working conditions. Associations between behavioral risk factors and general and mental health outcomes shown here and in previous studies [32,36,40,41,42,43] further warrant interventions to promote physical activity and maintenance of healthy weight among young employees, with a higher risk of poorer health.

One could also assume that occupational class contributes to the observed associations. It was derived from the employer’s register for those with an informed consent to use such register linkages, 83% of women and 80% of men [14], and, for the rest, it was derived from the survey. However, occupational class was not used as a predictor or covariate in the main final models due to its overlap with work exposures and thus the risk of over-adjustment The groups of managers and professionals, semi-professionals, routine non-manuals and manual workers reflect the type of work in terms of most likely exposures, distinguishing between having more likely physical (manual workers and routine-non manual) or psychosocial exposures (professionals and semi-professionals) at work. However, occupational class is not very suited to capture working conditions, as psychosocial exposures are present in lower level jobs, while physical work exposures are present in upper occupational classes, e.g., in the health care sector. As we have actual and rather specific reports of physical work exposures, we preferred to retain them in the models, since many other datasets only comprise class or focus on psychosocial working conditions. Moreover, some participants may not have completed their education, and their socioeconomic position is yet to be established. This further makes a focus on their working conditions more relevant than their current socioeconomic position. We used, nonetheless, parental education as an indicator of socioeconomic position. The effect of education was small (no further data shown).

In addition to self-rated health and common mental disorders, several other health outcomes could be of importance when focusing on the role of age at entry into paid employment and work exposures. For example, early exposures to physical workload factors (when aged 18–24 years) have been linked to low back disorders in midlife [44]. To confirm the importance of more detailed measures of physical work as compared to those previous studies, our additional analyses thus considered low back pain (LBP) as an outcome (prevalence of current LBP, 17%). However, of the examined adverse factors, only awkward working postures were associated with LBP. Nonetheless, the association between awkward working postures and LBP is in line with earlier prospective cohort studies [45,46] suggesting awkward working postures as risk factors for LBP. A prospective cohort study found that of physical workload factors, only awkward postures increase the risk of chronic LBP [45]. In the current study, we assessed the prevalence of current LBP, which is less prevalent than LBP in the past 6 or 12 months. Moreover, due to the nature of the study design, the participants with LBP may have reduced their exposure to workload factors or selected out of heavy physical work, or even out of paid employment. Alternatively, the lack of associations between other workload factors and LBP may indicate selection, where employees with LBP may have been among the non-respondents to the survey. An earlier study reported that those outside paid work are more likely to report sciatica [47], which could reflect such selection out of paid employment with heavy physical exposures, while suffering from low back disorders. Due to these issues and the specific nature of the measure, no further results of LBP are shown.

In our additional analyses regarding behavior-related risk factors as separate outcomes, we found that participants who are physically active during leisure-time report better SRH and are less likely to smoke or be with obesity. The findings are in line with earlier studies [48,49,50]. In these dose–response relationships, both moderate and high levels of leisure-time physical activity were inversely associated with SRH, smoking and obesity. The associations were stronger for high level of leisure time physical activity than for moderate level. Moreover, the current study supports the assumption that daily smokers are less engaged in leisure-time physical activity [51].

### 4.3. Methodological Considerations

A strength of this study is the use of a large sample of employees in their early and mid-careers, to study the significance of entry into paid employment to reports of health outcomes later, as well as contributions of adverse work exposures to adverse outcomes already in early and mid-career. The timing of entry into paid employment is very rarely included in large population surveys. Thus, an opportunity to focus on any sensitive periods or the role of age at entry is a strength and novelty of this study. Data on physical work exposures are also not collected in most large surveys, although physical work is still very common, particularly in lower occupational classes and within social and health care sector, with continuous physical exposure at work. Although working life has changed, physical exposures have not disappeared, and they still pose an important hazard to employee health. In addition to different physical work exposures, we could investigate sitting and standing as well as the number of hours doing physically heavy work per day. This helped produce a more concrete picture about the contributions of physical work to health of younger employees. The duration varied a lot between employees, and the results highlight that mostly those with two hours or more of physically demanding work per day could have the adverse health effects. Many jobs involve some physical exposures, so the results emphasize that it is important to focus on those jobs that involve mainly physical tasks. Another strength of our study was the use of validated and widely used measures for both predictors (working conditions) and health outcomes, such as SRH and CMD.

A limitation of this study is its cross-sectional nature. Entry into paid employment was retrospectively collected though, and most participants had had their first job already several years prior to entry into the study. Thus, they had been exposed to different physical work exposures already for some time and could have adverse health outcomes and behavior-related risk factors Regarding measurement of the risk factors, it is acknowledged that using dichotomous variables may have resulted in dilution of some of the effects. For example, due to small numbers and for practicality, current smokers were compared to all others. This means that ex-smokers and never-smokers were combined, and this could be misleading. Additionally, the study showed the links between employees’ work and health at one time point, which is also of importance, for example to plan targeted interventions, or enhance healthier behaviors and promote better health among those with most adverse exposures. As these employees are in their early career, and they should still have many decades of working life left, it is alarming how common poor mental and general health was, as well as adverse physical exposures at work. While an extensive non-response analysis established broad representativeness of the data regarding the target population [14], particularly manual workers and men were over-represented among the non-respondents. As physical work is particularly common in manual jobs, this suggests that we are likely to underestimate the prevalence of physical exposures in the cohort, and the results could be conservative.

## 5. Conclusions

Early entry into paid employment appears not to lead to poorer health during early work careers, but it may be linked to smoking and obesity. This held true after considering gender, age, socioeconomic factors and job strain. Furthermore, exposure to physically heavy work and awkward working postures are associated with adverse health outcomes already among younger employees. Working conditions are equally associated with poor health outcomes among those with and without behavior-related risk factors, thus preventive measures should be targeted early and followed up as the employees gain age. Improving working conditions already among young employees may help maintain later health and work ability.

## Figures and Tables

**Figure 1 ijerph-17-07854-f001:**
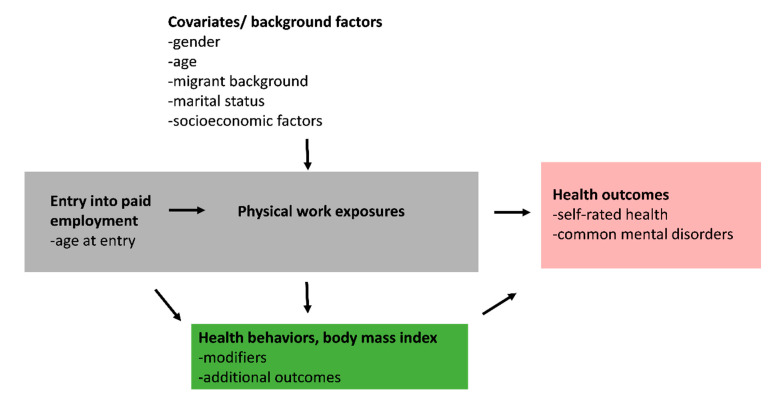
Study design and schematic framework of the study, presenting assumed associations between predictors (early entry into paid employment and physical work exposures), behavior-related risk factors (health behaviors and body mass index) and health outcomes.

**Table 1 ijerph-17-07854-t001:** The characteristics of the study population, proportions (%).

*Characteristic*	%
**Age at first employment**	
≤18	16.7
19–21	30.0
22–24	25.2
≥25	28.1
**Sociodemographic**	
Female	78.5
Migrant background (not born in Finland)	6.9
*Education*	
Elementary	1.0
Vocational	24.5
Upper secondary	10.3
Bachelor’s degree	35.5
Master’s degree	25.9
Doctoral degree	2.8
**Physical work exposures ^1^**	
Heavy physical exertion or lifting and carrying heavy loads	32.4
Uncomfortable working postures	55.8
Trunk rotation	39.4
Repetitive movements	39.7
Sitting	39.2
Standing	15.8
Vibration	4.9
**Behavior-related risk factors**	
*Body mass index*	
Healthy weight	57.3
Underweight	1.9
Overweight	26.9
Obesity	13.9
*Smoking*	
Past	20.4
Occasional	12.2
Current	11.3
*Binge drinking*	
Less than once a month	48.5
Once a month	17.5
At least once a week	6.4
**Health outcomes**	
Poor self-rated health	10.9
Common mental disorders	35.0

^1^ Each physical work exposure was as a separate item in the survey questionnaire. The table displays the percent of those reporting each of the exposure.

**Table 2 ijerph-17-07854-t002:** Odds ratios (OR) for the associations between age at first employment, physical work exposures, behavior-related risk factors and poor self-rated health and common mental disorders.

Characteristic	Age- and Gender-Adjusted Model				Multivariable Model			
	Poor Self-Rated Health		Common Mental Disorders		Poor Self-Rated Health		Common Mental Disorders	
	OR	95% CI	OR	95% CI	OR ^1^	95% CI	OR ^1^	95% CI
**Age at first employment ^2^** (ref: 22–24 years)								
≤18	1.22	0.94–1.59	1.09	0.90–1.31				
19–21	1.26	1.01–1.57	1.23	1.05–1.44				
≥25	0.78	0.61–0.99	1.13	0.96–1.34				
**Physical work exposures**								
Heavy physical exertion or lifting and carrying heavy loads	1.59	1.32–1.91	1.19	1.05–1.35	1.02	0.77–1.36	0.96	0.80–1.15
Uncomfortable working postures	1.92	1.58–2.33	1.38	1.23–1.56	1.15	0.86–1.53	1.22	1.02–1.45
Trunk rotation	1.70	1.42–2.03	1.16	1.03–1.31	1.01	0.74–1.38	0.89	0.73–1.09
Repetitive movements	1.83	1.53–2.20	1.38	1.23–1.56	1.24	0.95–1.61	1.18	0.99–1.40
Sitting	1.44	1.20–1.72	1.30	1.15–1.46	1.67	1.33–2.09	1.25	1.08–1.44
Standing	2.25	1.83–2.77	1.85	1.59–2.16	1.69	1.31–2.18	1.69	1.41–2.03
Vibration	1.39	0.95–2.03	1.14	0.87–1.51	0.90	0.58–1.41	0.92	0.67–1.25
**Behavior-related risk factors**								
Smoking (ref: never)								
Past	1.43	1.15–1.77	1.27	1.09–1.47	1.18	0.90–1.53	1.33	1.12–1.57
Occasional	1.32	1.01–1.72	1.27	1.06–1.53	1.13	0.82–1.57	1.21	0.99–1.49
Current	2.45	1.94–3.09	1.60	1.33–1.92	1.56	1.14–2.13	1.46	1.17–1.82
Body mass index (ref: healthy weight)								
Underweight	1.68	0.89–3.19	1.02	0.67–1.55	1.49	0.70–3.15	1.00	0.63–1.58
Overweight	2.15	1.75–2.65	1.03	0.89–1.18	1.94	1.50–2.50	0.92	0.79–1.08
Obesity	5.03	4.07–6.22	1.29	1.09–1.53	4.42	3.41–5.73	1.17	0.97–1.41
Leisure-time physical activity (tertile, ref: low)								
Moderate	0.48	0.39–0.60	0.93	0.81–1.08	0.58	0.46–0.74	1.00	0.86–1.18
High	0.36	0.28–0.45	0.84	0.73–0.97	0.50	0.39–0.65	0.94	0.80–1.10
Binge drinking (ref: never)								
Less than once a month	0.95	0.76–1.19	1.00	0.87–1.15	0.88	0.68–1.14	1.00	0.86–1.18
Once a month	1.19	0.90–1.58	1.45	1.21–1.74	1.00	0.72–1.40	1.36	1.10–1.67
At least once a week	1.89	1.32–2.70	1.71	1.31–2.22	1.44	0.94–2.19	1.58	1.18–2.12

^1^ Adjustment for age, gender, education, migrant background, household income, father’s education, mother’s education, job strain and for each other. ^2^ Age at first employment was not included in the final multivariable model because it was not associated with self-rated health or common mental disorders after adjustment for two or more variables.

**Table 3 ijerph-17-07854-t003:** Odds ratios (OR) for the associations between physical work exposures and health outcomes among participants with healthy weight or overweight or obesity.

Physical Work Exposure	Healthy Weight				Overweight or Obesity			
	Poor Self-Rated Health		Common Mental Disorders		Poor Self-Rated Health		Common Mental Disorders	
	OR ^1^	95% CI	OR ^1^	95% CI	OR ^1^	95% CI	OR ^1^	95% CI
Heavy physical exertion or lifting and carrying heavy loads	1.47	0.92–2.34	1.00	0.79–1.28	0.80	0.58–1.12	0.92	0.70–1.20
Uncomfortable working postures	0.86	0.54–1.38	1.19	0.95–1.49	1.41	0.99–1.99	1.18	0.90–1.55
Trunk rotation	0.93	0.55–1.55	0.89	0.68–1.15	0.94	0.65–1.35	0.79	0.59–1.07
Repetitive movements	1.11	0.70–1.74	1.20	0.95–1.50	1.17	0.86–1.61	1.13	0.88–1.45
Sitting	1.82	1.25–2.65	1.29	1.07–1.55	1.60	1.22–2.10	1.25	1.00–1.55
Standing	2.24	1.50–3.34	1.86	1.46–2.38	1.55	1.14–2.11	1.60	1.23–2.07
Vibration	0.99	0.46–2.12	0.74	0.47–1.17	0.88	0.53–1.46	1.09	0.72–1.66

^1^ Adjustment for age, gender, education, migrant background, household income, father’s education, mother’s education, job strain and for each other.

**Table 4 ijerph-17-07854-t004:** Odds ratios (OR) for the associations between physical work exposures and health outcomes among participants with low and high level of leisure-time activity ^1.^

Physical Work Exposure	Low Leisure-Time Activity				High Leisure-Time Activity			
	Poor Self-Rated Health		Common Mental Disorders		Poor Self-Rated Health		Common Mental Disorders	
	OR ^2^	95% CI	OR ^2^	95% CI	OR ^2^	95% CI	OR ^2^	95% CI
Heavy physical exertion or lifting and carrying heavy loads	0.97	0.71–1.31	0.97	0.78–1.20	1.34	0.78–2.31	0.98	0.72–1.33
Uncomfortable working postures	1.30	0.97–1.76	1.23	1.01–1.51	1.07	0.57–2.01	1.23	0.90–1.68
Trunk rotation	0.90	0.65–1.25	0.91	0.72–1.15	1.09	0.56–2.10	0.80	0.56–1.13
Repetitive movements	1.13	0.85–1.51	1.07	0.88–1.31	1.28	0.74–2.22	1.36	1.01–1.84
Sitting	1.65	1.29–2.11	1.22	1.03–1.45	1.99	1.26–3.14	1.34	1.04–1.71
Standing	1.90	1.45–2.49	1.62	1.31–2.01	1.33	0.80–2.20	1.96	1.43–2.68
Vibration	0.87	0.53–1.42	1.03	0.70–1.51	0.90	0.40–2.05	0.78	0.46–1.31

^1^ Median was used to dichotomize leisure time physical activity. ^2^ Adjustment for age, gender, education, migrant background, household income, father’s education, mother’s education, job strain and for each other.

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
