# Peer review of "Timing of Entry into Paid Employment, Adverse Physical Work Exposures and Health: The Young Helsinki Health Study"

_ijerph, 2020, doi:10.3390/ijerph17217854_

Round 1
Reviewer 1 Report
Report for
“Timing of entry into paid employment, adverse physical work exposures and health: The Young Helsinki Health Study”
This study examines whether timing of entry into paid employment and physical work exposures are associated with self-rated health and common mental disorders, and to determine whether behavior-related risk factors modify the associations.
This is a good study and the paper would provide a contribution to the literature. I will recommend publishing this paper in IJERPH. However, I have some comments on the paper.

Author Response
Reviewer 1
Report for
“Timing of entry into paid employment, adverse physical work exposures and health: The Young Helsinki Health Study”
This study examines whether timing of entry into paid employment and physical work exposures are associated with self-rated health and common mental disorders, and to determine whether behavior-related risk factors modify the associations.
This is a good study and the paper would provide a contribution to the literature. I will recommend publishing this paper in IJERPH. However, I have some comments on the paper.
- In Section 2.1, I would like to see the numbers (or proportions) of municipal employ in target population and in general.
RESPONSE: Thank you for your positive and clear feedback. We have added the proportions in the target population, as requested. The following sentence was added to section 2.1: “Thus, in the target population, the proportion of women is 77%, and that of men 23%, while the corresponding figures among respondents were 79% and 21%, respectively.” In the Finnish municipal sector in general, across all age groups, 80% of employees are women. More information about the municipal sector in Finland can be found here: https://www.kt.fi/en/municipal-sector-and-personnel.
- Section 2.2, three age groups and four levels?
RESPONSE: Three age groups was a mistake and it has been now corrected. Thank you for noticing this (please see section “Predictors: timing of entry into paid employment and physical work exposures”, first paragraph).
- Potential modifiers: Why consider current smokers (only 11%) vs all other? Why not never smoked vs all other? Also, what do you mean by small number of smokers, and number for binge drinking were not sufficient?
Also, it is not clear what do authors mean by ``Leisure time physical activity was first dichotomize into inactive and active’’? Then following sentence reads `` the median value of MET was used as cut point to dichotomize leisure time physical activity? Two different explanations.
RESPONSE: When we conducted stratified analyses, the numbers did not allow for more categories in these health behaviors. Hence, we only used these dichotomies. We agree though that it would be interesting to distinguish between never and ex-smokers. We have clarified the formation of the physical activity variable. It was dichotomized (inactive and active), and we used the median value of MET (metabolic equivalent values were used to measure physical activity) in the dichotomization, to distinguish between inactive and active participants. Thus, the sentences are not different but the second one was meant to provide more information about how the dichotomization was done. We hope the current version seat the end of the section “Potential modifiers of the associations: health behaviors and obesity”: Leisure-time physical activity was first dichotomized into inactive and active, using the median value of MET as a cut point.
- Figure 1 should be place under ``Statistical analysis`` section.
RESPONSE: We have replaced the Figure 1 under the Statistical analyses, as suggested.
- Table 1: The age category >=25 is ambiguous, shouldn’t it be 25-39??
RESPONSE: We appreciate the comment. However, we have used the >25 as the last category, as this is not age distribution but about age when first entering paid employment. We think it is clearer to distinguish between earlier and later entry into paid employment, not highlighting the minimum and maximum values. Although the oldest participants were 39 years when they returned the survey, this does not reflect their age at entry into paid employment. If we added the earliest and latest possible entry age, they could highlight an outlier.
- Section 3.1: Mean age =32 seems bit odd because only 28% falls in age group over 25. Majority of the respondent (about 55%) are in age group 19-24. Need to re-check.
RESPONSE: In Table 1, as described above, we are actually not presenting the prevalence of age groups but the distribution when these participants first entered paid employment. Thus, 28.1% had their first job when they were 25 years old or older. The mean age of the participants when they participated the survey was, in turn, 32 years. The information about their age at entry into paid employment was collected retrospectively and they were all employed when invited to the study. We have checked that the manuscript is clear in this respect, and reference to Table 1 is given in the results section only after mentioning mean age when completing the questionnaire.
- Table 1: It seems like a person can tick more than one option to ``Physical work exposure’’ (check proportion adds to over 100%). This need to be explained/discussed in text.
RESPONSE: We have clarified the measurement of physical work exposures. The proportions add to over 100% because these were all different items in the survey. Thus, the proportions show who had each exposure (vs. who did not). In other words, each row adds to 100% but we are displaying only the proportion of employees having the exposure. This is also true in real life, as many people have different types of physical workload factors and exposures in their job. The following footnote was added to help interpret the Table 1:
1 Each physical work exposure was a separate item in the survey questionnaire. The table displays % of those reporting each of the exposure.
- BMI has four categories, but Table 1 presents only two categories?
RESPONSE: Thank you for noticing that. We have added the categories of underweight and healthy weight in the Table 1. In the stratified analyses, we used a dichotomy (overweight/obese vs. others), as it is not feasible to do the models in more detailed groups. Additionally, in stratified analysis the small group (1.9%) of underweight participants was excluded from the analysis.
- Table 2: Why age group 22-24 considered as reference category? Also, age group over 25 is also statistically significant but not discussed in result section.
RESPONSE: It was used as a reference as in Finland it is typical to enter first employment at that age. As educational systems notably differ between countries, it is not meaningful to use other countries’ typical age at entry as a reference. Education is typically quite long in Finland, and most people graduate from (general) upper secondary education when they are 18 to 19 years old. After that there is higher education which often takes 3 to 5 years (or more). However, around 23 years is considered a typical age, when one can have completed education. It is of note that education in Finland is free for all, including universities. More information about the Finnish educational system can be found from here: https://minedu.fi/en/education-system and in more detail here: https://www.oph.fi/en/education-system.
If entry to paid employment is very early it could mean socioeconomic disadvantage in the family, a need to fund the education or inability to get into higher education (ref 14). A late entry is also not desired, considering the efforts to extend working life and it could also mean e.g. periods of unemployment or health or other problems. There is a proportion of young adults, who have not had an entry into paid employment but have long-term unemployment (ref 15). However, these are just proxies and groups are likely to be very heterogeneous. Anyway, the idea is to distinguish between too early and too late entries, which both could reflect some issues. These questions have not been widely studied so there are few suitable references to add. This is one of the reasons we aimed to study these questions, to establish if age at entry has an effect on health among young employees, which could in turn mean later risk for work disability and predict unstable work careers. More research is needed here, but it is beyond this already rather extensive, yet explorative study.
We have clarified the section 2.2 to briefly refer to our educational system.
- Page 8: Authors mentioned only current smoking associated with SRH, while from Table 2, all smoking categories are associated with SRH. Need to re-check and re-write. Same with BMI association with SRH. Also check association between smoking categories and CMD.
RESPONSE: The text in the results focused on multivariable models, and after full adjustments, only current smoking was associated with SRH. It is true there were associations after adjusting for gender and age, but to keep the results section more brief, we preferred not to repeat all the associations from the tables but instead focus on the adjusted ones, particularly for behavioral risk factors that were not the main focus of this study. We have clarified the section to repeat we refer to multivariable models (see page 8: “In multivariable models, current smoking, overweight and obesity, were positively associated with poor SRH, while moderate and high levels of leisure-time physical activity were inversely associated with poor SRH.”), and we have also clarified other sentences to include more details.
- Section 3.3: What do you mean by ``goodness of fit was poor’’?
RESPONSE: We have clarified the section 3.3 that we mean the model fit. This refers to the fact that adding the behavioral risk factors in the models resulted in poorer fit of the model, and hence they were not included in the final model.
- I can not access to supplementary Tables. Link provided in manuscript is broken.
RESPONSE: The link may not be functional yet, and could be pending the acceptance of the paper. Supplementary tables were submitted as a separate file. We hope that they are visible and accessible in this revised version.
- Additional Analysis: The sentence ``inverse association between binge drinking and poor SRH in men’’ need to re-check. It doesn’t look correct.
Also, in paragraph 5, a sentence `` more likely to smoke and be obese’’, should not it be ``less likely to smoke and be obese’’? These participants are more likely to involve in Physical activity, and it doesn’t make sense that these participants are more likely to be obese.
RESPONSE: We have re-checked the results regarding the association between binge drinking and self-rated health among men. They are correct. In the age-adjusted model, the ORs appeared to be below 1 among men, while in the main analyses, dominated by women (Table 2), the association is as expected, with binge drinkers having poorer self-rated health. Binge drinking is more common among men than women, but the number of men is a lot smaller, and a non-significant ‘inverse association’ should therefore not be over-interpreted as it could be due to mere chance. It is of note that the confidence intervals were wide and included 1.00: OR 0.66; 95% CI 0.35-1.27 for binge drinking once a week. We have revised the text not to highlight the inverse association (please see 3.4, first paragraph).
- Comparing results of current smokers with all other could be miss-leading because only 11% represents current smokers.
RESPONSE: It is true that the reference group also comprised ex-smokers, which could dilute the associations. In many studies, a dichotomy is used, e.g. as here for feasibility. We have acknowledged this issue in the revised discussion (last paragraph of Methodological considerations: “Regarding measurement of the risk factors, it is acknowledged that using dichotomous variables may have resulted in dilution of some of the effects. For example, due to small numbers and for practicality, current smokers were compared to all others. This means that ex-smokers and never-smokers were combined, and this could be misleading.”).
- Abbreviations: missing CMD and LBP.
RESPONSE: We have added the missing abbreviations (please see the corresponding section ‘Abbreviations’).
Reviewer 2 Report
The aim of this study was to determine the associations of age at entry into paid employment and physical work exposures with general and mental health in young employees and determined whether associations differ by behavior-related risk factors. Overall, the paper is well-written, and the methods and results of the study were very detailed, though there were much fewer male participants to effectively compare the differences between men and women in this study.
However, perhaps in the final version of the paper, consider taking out or shortened the section on 3.4 Additional analyses for enhanced readability of the paper.
Author Response
Reviewer 2
Comments and Suggestions for Authors
The aim of this study was to determine the associations of age at entry into paid employment and physical work exposures with general and mental health in young employees and determined whether associations differ by behavior-related risk factors. Overall, the paper is well-written, and the methods and results of the study were very detailed, though there were much fewer male participants to effectively compare the differences between men and women in this study.
However, perhaps in the final version of the paper, consider taking out or shortened the section on 3.4 Additional analyses for enhanced readability of the paper.
RESPONSE: Thank you for these very positive comments. We have shortened the section 3.4, as suggested, to improve readability.

Reviewer 3 Report
This is an interesting well-written study on the health effects of the timing of entry into paid employment and adverse physical work exposures in a young cohort in Helsinki. This manuscript is under the scope and suitable to be publish in the IJERPH after MINOR REVISION. Some specific comments are provided below: Authors did not provide any information about the study's ethic approval or participant’s informed consent. More importantly, extensive self-citations and repetitions must be avoided throughout the text. In page 3, authors affirmed that "age at entry was classified into three groups distinguishing between early, average and late entry into first paid employment: 25 years, respectively" but provided 4 age groups, including a reference. Why the average group was not used as reference? Please explain better. In page 3-4, authors stated that "As an outcome it was dichotomized to distinguish participants with obesity from others. In the stratified analyses, it was dichotomized to distinguish participants with overweight or obesity from others." Why this changed between these analysis? In page 12, authors declared that: “In turn, physical work exposures were associated with self-rated and mental health”. In my opinion, more information about what work exposures were significantly associated with the health effects should be provided. Moreover, the statement “As physical work is still prevalent within the public sector, these results thus highlight the importance of early detection of risk groups (…)” seems to reduce the problem of physical work to public sector, what is not true. At the end of page 12: “Moreover, some participants may not (yet) have completed their education, and their socioeconomic position is yet to be established. This further makes a focus on their working conditions more relevant than their current socioeconomic position. We used, nonetheless, parental education as an indicator of socioeconomic position.” Is there any external reference to support this decision? In the limitations, authors did not mention the non-probabilistic convenience sampling, which could potentially result in a selection bias. In conclusions, authors affirmed “Early entry into paid employment appears not to lead to poorer health during early work careers, but it may be linked to smoking and obesity.” They should specify if that happened for all workers, independent on their sex or working conditions, for instance.Author Response
Reviewer 3
Comments and Suggestions for Authors
This is an interesting well-written study on the health effects of the timing of entry into paid employment and adverse physical work exposures in a young cohort in Helsinki. This manuscript is under the scope and suitable to be publish in the IJERPH after MINOR REVISION. Some specific comments are provided below:
Authors did not provide any information about the study's ethic approval or participant’s informed consent.
RESPONSE: Thank you for these positive comments. We have added a comment about ethical approval of this study and informed consent (please see end of the section 2.1). All participants were given detailed information about the study according to the GDPR, and then they either chose to return to survey or not.
More importantly, extensive self-citations and repetitions must be avoided throughout the text.
RESPONSE: We have revised our references to cut the numbers of self-citations. It is true that there were many citations by the authors. These studies, however, were not conducted in the current cohort but used an earlier cohort of older employees, as well as e.g. two different birth cohorts in Finland. The current data are relatively new, and only a non-response analysis has been published so far. Moreover, a very few large cohorts include data on physical working conditions and these outcomes among young employees, and also some cohorts may not be comparable to this one. Therefore, where applicable and available, we have cited the studies within the area but not conducted by us. However, overall, the citations were selected to cover similar measures and more comparable occupational cohorts.
In page 3, authors affirmed that "age at entry was classified into three groups distinguishing between early, average and late entry into first paid employment: 25 years, respectively" but provided 4 age groups, including a reference. Why the average group was not used as reference? Please explain better.
RESPONSE: This was a mistake, thank you for noticing it. It is true that there were four groups. This has been corrected, please see section “Predictors: timing of entry into paid employment and physical work exposures”, first paragraph: “…age at entry was classified into four groups distinguishing between early, average and late entry into first paid employment: <18 years, 19–21 years, 22–24 years (reference) and >25 years, respectively”.
In page 3-4, authors stated that "As an outcome it was dichotomized to distinguish participants with obesity from others. In the stratified analyses, it was dichotomized to distinguish participants with overweight or obesity from others." Why this changed between these analysis?
RESPONSE: In the stratified analyses, BMI was dichotomized to distinguish participants with overweight or obesity from others because small numbers of people with obesity did not allow for stratified analyses to be conducted. We have clarified this in the revised manuscript. However, obesity worked as an outcome, and it was used as it is known to be a higher risk factor as compared to overweight.
In page 12, authors declared that: “In turn, physical work exposures were associated with self-rated and mental health”. In my opinion, more information about what work exposures were significantly associated with the health effects should be provided.
RESPONSE: The sentence was in the beginning of the discussion, as part of the main findings, which aimed to very briefly summarize the results that were presented in previous sections and tables. We prefer to keep the main findings section brief and discuss the main findings without too many details and avoiding repetition with the results. We have, however, added an example to be clearer, please see page 12: “In turn, physical work exposures were associated with self-rated and mental health particularly before full adjustments, and sitting and standing after full adjustments.”
Moreover, the statement “As physical work is still prevalent within the public sector, these results thus highlight the importance of early detection of risk groups (…)” seems to reduce the problem of physical work to public sector, what is not true.
RESPONSE: We have revised the sentence to avoid giving a wrong impression. We fully agree that physical work is present also outside the public (municipal) sector. The intention was only to highlight that it is very common within public sector and among women as well, even though many previous studies focus on psychosocial work (Please see revised Discussion, section: Interpretation, 1st paragraph: “As physical work is still prevalent, these results thus highlight the importance of early detection of risk groups, to promote health and prevent the health problems to become chronic.”).
At the end of page 12: “Moreover, some participants may not (yet) have completed their education, and their socioeconomic position is yet to be established. This further makes a focus on their working conditions more relevant than their current socioeconomic position. We used, nonetheless, parental education as an indicator of socioeconomic position.” Is there any external reference to support this decision?
RESPONSE: This was based on the Finnish system, not so much on previous studies, as educational systems largely vary between countries. As we focus on employees aged 18 to 39, many of the younger employees could still continue in higher education, and they may, for example, be working before entering university or also during their education. As not all had completed their education at the time of the survey, their socioeconomic position is likely to change. Thus, we chose to focus on working conditions. Occupational class, as an indicator of socioeconomic position, is strongly correlated with working conditions and thus adjusting for it would not be meaningful.
We have clarified the section 2.2 about education. More information of the Finnish educational system can be found from here: https://minedu.fi/en/education-system and in more detail here: https://www.oph.fi/en/education-system.
In the limitations, authors did not mention the non-probabilistic convenience sampling, which could potentially result in a selection bias.
RESPONSE: This actually was not a non-probabilistic convenience sample. We have also published a non-response analysis cited in the paper, where we describe the representativeness of the data as compared to the target population. In non-probabilistic convenience samples, there is no such possibility.
In conclusions, authors affirmed “Early entry into paid employment appears not to lead to poorer health during early work careers, but it may be linked to smoking and obesity.” They should specify if that happened for all workers, independent on their sex or working conditions, for instance.
RESPONSE: The main conclusion is that the associations between early entry and behavioral risk factors remained in multivariable models. We have clarified the conclusions, as suggested (“Early entry into paid employment appears not to lead to poorer health during early work careers, but it may be linked to smoking and obesity. This held true after controlling for gender, age, socioeconomic factors, and job strain.”).
Our main analyses were done in the pooled data, as there were no interactions by gender and due to small numbers of men. Thus, the results for men should be confirmed in larger studies. We have added this limitation to the Methodological considerations section on page 14: “A limitation of this study is its cross-sectional nature. Moreover, as the majority of the respondents were female, results for men need to be confirmed in other larger studies.”